# Targeted Radionuclide Therapy of Prostate Cancer—From Basic Research to Clinical Perspectives

**DOI:** 10.3390/molecules25071743

**Published:** 2020-04-10

**Authors:** Malwina Czerwińska, Aleksander Bilewicz, Marcin Kruszewski, Aneta Wegierek-Ciuk, Anna Lankoff

**Affiliations:** 1Centre for Radiobiology and Biological Dosimetry, Institute of Nuclear Chemistry and Technology, Dorodna 16, 03-195 Warsaw, Poland; m.wasilewska@ichtj.waw.pl (M.C.); m.kruszewski@ichtj.waw.pl (M.K.); 2Centre of Radiochemistry and Nuclear Chemistry, Institute of Nuclear Chemistry and Technology, Dorodna 16, 03-195 Warsaw, Poland; a.bilewicz@ichtj.waw.pl; 3Department of Molecular Biology and Translational Research, Institute of Rural Health, Jaczewskiego 2, 20-090 Lublin, Poland; 4Department of Medical Biology, Institute of Biology, Jan Kochanowski University, Uniwersytecka 7, 24-406 Kielce, Poland; aneta.wegierek-ciuk@ujk.edu.pl

**Keywords:** prostate targeted therapy, prostate cell-surface receptors, PSMA ligands, PSMA-targeted radioimmunoconjugates

## Abstract

Prostate cancer is the most commonly diagnosed malignancy in men and the second leading cause of cancer-related deaths in Western civilization. Although localized prostate cancer can be treated effectively in different ways, almost all patients progress to the incurable metastatic castration-resistant prostate cancer. Due to the significant mortality and morbidity rate associated with the progression of this disease, there is an urgent need for new and targeted treatments. In this review, we summarize the recent advances in research on identification of prostate tissue-specific antigens for targeted therapy, generation of highly specific and selective molecules targeting these antigens, availability of therapeutic radionuclides for widespread medical applications, and recent achievements in the development of new-generation small-molecule inhibitors and antibody-based strategies for targeted prostate cancer therapy with alpha-, beta-, and Auger electron-emitting radionuclides.

## 1. Introduction

According to the cancer epidemiology databases provided by the International Agency for Research on Cancer and the WHO Cancer Mortality Database, prostate cancer is the most commonly diagnosed malignancy in men and the second leading cause of cancer-related deaths in Western civilization [1]. Today, standard primary therapy for patients with localized prostate cancer consists mainly of radical prostatectomy and/or external beam radiotherapy or brachytherapy. In the case of recurrent disease or advanced-stage prostate cancer, the main therapy is androgen ablation using luteinizing hormone releasing hormone (LHRH) agonists and antagonists and/or anti-androgen receptors (ARs) [2,3]. Although localized prostate cancer can be treated effectively by these therapies, almost all patients ultimately progress to metastatic castration-resistant prostate cancer (mCRPC) [4]. Most patients with metastatic disease initially respond to androgen deprivation therapy, taxane-based chemotherapies, immunotherapy, or radium-223, but each of these regimens provides only limited 2–4 months median survival benefit [5,6]. The median survival for men with mCRPC ranges from 13–32 months with a 15% 5-year survival rate. Most deaths from prostate cancer are attributed to the incurable, late stage cancer form [7,8]. Due to the significant mortality and morbidity rate associated with the progression of this disease, there is an urgent need for new and targeted treatments. Prostate cancer is an excellent target for targeted therapies for several reasons: (*1*) the prostate is a nonvital organ, thereby allowing targeting of tissue-specific antigens, (*2*) prostate cancer metastases predominantly involve lymph nodes and bones, locations that receive high levels of circulating antibodies, (*3*) prostate cancer metastases are typically small in volume, allowing good antibody access and penetration, and (*4*) the prostate-specific antigen (PSA) serum marker provides a means for the early detection of metastases and the monitoring of therapeutic efficacy [9]. In this review, we summarize the recent advances in research on identification of prostate tissue-specific antigens, generation of highly specific and selective molecules targeting these antigens, availability of therapeutic radioisotopes for widespread medical applications and recent achievements in the development of a new generation of antibody-based strategies and small-molecule inhibitors for targeted prostate cancer therapy with alpha- and beta-particle radionuclides.

## 2. Potential Therapy Targets for Prostate Cancer

The identification of potential therapy targets in advanced prostate cancer and androgen-independent disease is critical for improving diagnosis and therapy. Ideal targets for prostate cancer therapy would include structures that are exclusively expressed in normal prostate tissue, which are highly expressed in metastatic disease, and that are accessible to therapeutic modalities at the cell surface. So far, several cell-surface proteins, glycoproteins, receptors, enzymes and peptides have been tested as targets for the treatment of prostate cancer [10,11,12] (Figure 1).

### 2.1. Prostate-Specific Membrane Antigen (PSMA)

One of the most important membrane antigens foreseen for targeted therapy is prostate-specific membrane antigen (PSMA), also known as *N*-acetyl-alpha-linked acidic dipeptidase I (NAALA-Dase), glutamate carboxy-peptidase II (EC 3.4.17.21) or folate hydrolase. PSMA is a type II membrane glycoprotein of about 100 kDa, with a short intracellular domain of 19 amino acids, a transmembrane domain of 24 amino acids, and an extensive glycosylated extracellular domain of 707 amino acids. The extracellular domain has been crystallized and its structure revealed a symmetric dimer with each polypeptide chain containing three domains analogous to the three TfR1 domains: a protease domain, an apical domain, and a helical domain. A large cavity at the interface between the three domains includes a binuclear zinc site and predominantly polar residues. The observation of two zinc ions and conservation of many of the cavity-forming residues among PSMA orthologs and homologs identify the cavity as the substrate binding site [13]. PSMA is highly expressed in prostate cells. The expression of PSMA is up-regulated in malignant disease, with the highest level detected in metastatic androgen-independent prostate cancer. PSMA is also over-expressed in the neovascular endothelium of most solid tumors, including lung, colon, breast, renal, transitional cell, and pancreas cancers, but it is not expressed in normal vasculature [14]. PSMA is also expressed in other tissues including normal (benign) prostate epithelium, in female Skene’s glands, small intestine, renal tubular cells, and salivary gland; however, this “non-target” expression is 100–1000-fold less than baseline expression in prostate cancer [15,16]. Furthermore, the sites of PSMA expression within these organs are not typically exposed to circulating antibodies which generally do not cross intact basement membrane and tight junctions required to access these sites of non-PC PSMA expression. The PSMA undergoes receptor-mediated endocytosis. Analysis by immunofluorescence and electron microscopy techniques has proven that PSMA undergoes internalization through clathrin-coated pits to be transported into lysosomes [17]. Moreover, the internalization rate is enhanced up to three-fold in a dose-dependent manner by anti-PSMA antibody (mAb) binding. To summarize, this characteristic makes PSMA excellent target of prostate cancer, because it is (*1*) primarily expressed in the prostate, (*2*) abundantly expressed as protein at all stages of the disease, (*3*) presented at the cell surface but not released into the circulation, (*4*) up-regulated in metastatic disease, (*5*) associated with enzymatic activity, and (*7*) internalized after antibody binding by receptor-mediated endocytosis. These findings have spurred the development of many potential ligands to this enzyme for single photon emission computed tomography (SPECT) imaging, positron emission tomography (PET) and prostate cancer therapy [18,19,20,21,22,23] (see Section 3).

### 2.2. Prostate Stem Cell Antigen (PSCA)

Prostate stem cell antigen (PSCA) is a glycosylphosphatidylinositol-anchored cell-surface protein belonging to the Ly-6/Thy-1 family of cell-surface antigens, initially identified in a human prostate cancer cell xenograft [24]. PSCA is expressed in some normal tissues, such as the prostate, bladder, neuroendocrine cells of the stomach, colon, and brain, but its expression in prostate cancer and other cancer tissues is significantly higher compared to normal tissues [11,25,26]. Recent studies revealed that PSCA is over-expressed in 83%–100% of prostate cancers, in the great majority of prostate cancer bone metastases (87%–100%) and in many metastases to other sites (67% liver, 67%–95% lymph node) [27,28,29] Heinrich et al., 2018). The higher level of PSCA expression is directly associated with the advanced stages, high Gleason score, local and prostatic capsular invasion and androgen independency of the disease [30], raising the possibility that PSCA may have diagnostic and therapeutic applications in prostate cancer [31]. Preclinical in vitro and in vivo studies, evaluating the therapeutic efficacy of anti-PSCA mAbs [1G8 (IgG1kappa) and 3C5 (IgG2akappa) in-human prostate cancer xenograft mouse models showed a significantly prolonged survival and inhibition of metastatic tumor formation with anti-PSCA mAb treatment [32,33]. In a slightly different strategy, an anti-PSCA mAb conjugated to maytansinoid DM1 (a potent antimicrotubule agent) caused complete regression of established tumors in a large proportion of animals [26]. Studies with dual docetaxel/superparamagnetic iron oxide loaded nanoparticles modified with PSCA-specific ScFv, the single-chain PSCA antibody scAb-PSCA or anti-PSCA mAbs (1G8 and 3C5 59) demonstrated prostate cancer-specific accumulation in vitro and in vivo [26,34,35]. In 2012, the first in-human phase I/IIA study was designed to evaluate the safety and pharmacokinetics of AGS-PSCA (a fully human monoclonal antibody directed to PSCA) in progressive castration-resistant prostate cancer [36]. This study demonstrated the safety of using a monoclonal antibody directed against PSCA. However, the results revealed also that anti-tumor effects using a naked antibody were limited in this trial. Studies with radioactive ligands targeted to PSCA analogues have also been explored. These studies reported that hu1G8 minibody (a humanized anti-PSCA antibody fragment (single-chain Fv-CH3 dimer) and hu1G8 antibody fragments (A2, A11, and C5) radiolabeled with the positron emitter ^124^I demonstrated improved sensitivity and specificity of clinical prostate cancer metastasis detection over bone scans, which are the current clinical standard of care [37,38,39,40]. Yu et al. [41] tested the potential therapeutic efficacy of mouse anti-human PSCA IgG monoclonal antibodies labeled with ^131^I for prostate cancer therapy. These studies revealed that ^131^I-PSCA mAb showed great advantages in radioimmunotherapy and stronger positive anti-prostate tumor activity compared with ^131^I-IgG and PSCA mAb in tumor-bearing nude mice.

### 2.3. Six-Transmembrane Epithelial Antigen of the Prostate (STEAP)

The 6-transmembrane epithelial antigen of prostate (STEAP) family of proteins consists of 4 members, named 6-transmembrane epithelial antigen of prostate 1 to 4 (STEAP1–STEAP4) [42]. They all have a C-terminal 6-transmembrane domain composed of six-transmembrane helices with distant homology to yeast ferric metalloreductases and to mammalian dihydronicotinamide-adenine dinucleotide phosphate (NADPH) oxidase, and an N-terminal with homology to the archaeal and bacterial F420H2: NADPþ oxidoreductase (FNO)–binding proteins [43]. A body of literature demonstrates altered STEAP gene expression patterns within several cancers, suggesting the STEAP family members as important targets for a targeted cancer therapy.

STEAP1 acts as an ion channel or transporter protein in tight junctions, gap junctions, or in cell adhesion, taking part in intercellular communication [44]. STEAP1 is over-expressed in many cancer cells, where promotes proliferation and prevent apoptosis [45]. STEAP1 over-expression in prostate cancer and its bone metastases has been very well documented, showing correlation between increased expression and tumor aggressiveness [46]. Expression of STEAP1 has been also reported in normal prostate epithelium, pericardium, peritoneum, fetal and adult liver, as well as human umbilical vein endothelial cells, but it is markedly lower as compared to cancer tissues [47]. A phase I study of the DSTP3086S (the humanized IgG1 anti-STEAP1 monoclonal antibody linked to the potent anti-mitotic agent - monomethyl auristatin E (MMAE) in patients with mCRPC showed its good tolerability and anti-tumor activity [48].

STEAP2 is involved in the endocytotic and secretory pathways [49]. It acts also as a ferrireductase and cupric reductase, which stimulates the uptake of iron and copper into the cell [50]. In vitro and in vivo studies revealed that STEAP2 regulates several genes involved in the cell cycle, increases cancer cell proliferation and results in increased migration and invasion of normal prostate epithelial cells, suggesting a possible role in prostate carcinogenesis [51]. STEAP2 has been detected in androgen receptor (AR)-negative and (AR)-sensitive prostate cancer cell lines [52,53]. Moreover, recent evidence has demonstrated up-regulation of STEAP2 protein within prostate cancers compared to normal glands [54].

STEAP3 reduces endosomal ferric iron bound to transferrin to the ferrous form and facilitates exosome secretion via a non-classic pathway [50,55]. It is strongly expressed in prostate cancer, bone marrow, and fetal liver, whereas cell-surface expression in normal tissues is restricted to prostate, liver, dorsal root ganglia and bladder [44]. Recently, the STEAP3 gene has been reported to be expressed in glioblastoma cancer [56].

STEAP4 is involved in iron and copper homeostasis as well as in secretory/endocytic pathways [57,58]. In situ hybridization of human prostate cancer specimens showed that STEAP4 is expressed in epithelial prostate cells, and that its expression is significantly higher in prostate tumors compared with normal glands, suggesting that they may play a role in prostate cancer development and progression [49]. However, the increased STEAP4 expression level has also been observed in nonneuronal tissues with relatively high levels in pericardium, synovial membranes, placenta, adipose tissue, lung, heart and liver [47].

In conclusion, strong expression of STEAPs receptors in prostate cancer and their localization on the cell surface suggest that STEAPs could be promising targets for targeted prostate cancer therapy and diagnosis. Nevertheless, due to their relatively high expression in normal tissues, the value of these receptors as an attractive therapeutic target is questionable.

### 2.4. G-Protein Coupled Receptor (5-oxoER)

G-protein coupled receptor (5-oxoER) belongs to the G-protein-coupled receptor (GPCR) superfamily, which is characterized by seven transmembrane domains with an extracellular N-terminus and a cytoplasmic C-terminus [59]. GPCRs respond to a variety of extracellular signals such as peptides, hormone proteins, proteases, biogenic amines, nucleotides, and lipids. The receptors couple with heterotrimeric G proteins to transduce their signal across the membrane and into the cell. They are divided into distinct subfamilies according to the types of ligands and sequence homology. Among them, the G-protein-coupled oxoeicosanoid 5-oxoETE receptor (OXER1, 5-oxoER, 5-oxo-6E,8Z,11Z,14Z) presents prostate tumor associated over-expression [60,61,62]. OXER1 activates PKC-epsilon via phospholipase C-beta (PLC-beta), which generates diacylglycerol (DAG) to activate PKC-epsilon [63]. More recently, a detailed analysis of the Cancer Genome Atlas entries confirmed over-expression of OXER1 protein in prostate cancer tissues. However, this expression was observed also in normal cells and various types of human cancer cells lines, including breast, lung, ovaries, colon, and pancreas [64].

## 3. PSMA Ligands for Targeted Radiotherapy

Through the past decade, a variety of PSMA-targeted ligands has been systematically evaluated for their modification to radiopharmaceuticals [65]. Among them, two distinct approaches have been used for targeting PSMA, the first approach taking advantage of the macromolecular protein structure of PSMA to provide specific monoclonal antibodies and antibody-based molecules as targeting vectors [66]. The second approach takes advantage of the enzyme activity of PSMA and uses small-molecule enzyme inhibitors or binding agents as target seeking agents [67].

### 3.1. Antibodies

Antibodies are proteins (~150 kDa; ~20 nm) composed of two identical heavy chains (Hc) (~50 kDa) and two light chains (Lc) (~25 kDa). The heavy and light chains are held together by S–S bonds forming a Y-shape consisting of constant domains (CH and CL) and variable domains (VH and VL). The two VH and VL domains are highly specific for the target antigen forming the antigen-binding fragment (Fab) [68]. The first published antibody against PSMA was the murine mAb 7E11-C5.3 (capromab) [69]. This antibody was used for the initial validation of PSMA in vivo. Molecular mapping revealed that 7E11-C5.3 only targets the intracellular domain, which is not exposed on the cell surface, and therefore cannot bind to viable cells [70]. These limitations were mitigated in 1997 by the development of second generation mAbs (J591, 5533, 5415, E99), specific to the extracellular epitopes of human PSMA [71]. In the following years great efforts were undertaken to develop mAbs against the extracellular domain of PSMA. Summarizing these efforts, numerous papers describing in detail the mAbs different characteristics have been published [72,73,74,75,76,77]. Of these, the mAbs J591, J533, J415, D2B, 107-1A4, E99, 3/A12, 3/E7, 3/F11 and 3E6 have been shown to bind the most efficiently to cell-surface PSMA. To date, several in vitro and in vivo targeted therapy approaches were investigated with these antibodies and the first-generation products have entered clinical testing [78,79]. Among them anti-PSMA-based vaccine approaches have been tested and immune responses have been demonstrated in the absence of significant toxicity [80]. Therapy with drug-conjugated and immunotoxin-conjugated antibodies has yielded objective clinical responses as measured by reductions in serum prostate-specific antigen and/or tumor volume. However, responses were observed in a minor fraction of patients and at doses near the maximum tolerated dose [81]. These obstacles can be overcome by designing anti-PSMA radio-immunoconjugates for targeted radionuclide therapy (see Section 3).

### 3.2. Minibodies, Diabodies, Single-Chain Variable Region Fragments, and Nanobodies

Though antibodies possess features that favor them to achieve high tumor uptake, their large size hampers an access to tumor cells and their tumor penetration capabilities [82]. To overcome this problem, smaller PSMA antibody fragments have been developed, including minibodies, diabodies, single-chain variable region fragments, nanobodies, and monobodies (Figure 2).

Minibodies (~80 kDa) are bivalent homodimers, with each monomer containing the variable domains scFv linked to the human IgG1 hinge and a CH3 domain (dimeric scFv-CH3) [83]. PSMA-targeted minibodies were genetically engineered from the huJ591 antibody and their feasibility was explored as PET imaging probe with ^89^Zr as the radiometal of choice [84]. The first in-human phase I dose-escalation study with ^89^Zr-desferrioxamine-IAB2M minibody (^89^Zr-IAB2M) was conducted in patients with metastatic prostate cancer [85]. Phase I/IIa PET imaging study with ^89^Zr conjugated with desferroxamine-IAB2M (^89^Zr-Df-IAB2M) minibody was performed to assess its safety and feasibility in patients with urological cancer [86].

Diabodies (~50 kDa) are bivalent homodimeric (monospecific) or heterodimeric (bispecific) antibody fragments constructed by connecting only heavy chain variable fragments (VH) and light-chain variable fragments (VL) with a flexible linker of 5∼8 amino acid [87]. They penetrate cancer tissue more effectively than conventional antibodies and are likely to represent a good balance between circulation time/systemic clearance, target accumulation, and tissue penetration. PSMA-targeted diabodies were developed from the huJ591 and D2B antibodies (PSMA-targeted huJ591 cysteine-modified scFv diabody labeled with ^89^Zr, PSMA-targeted cysteine-modified scFv diabody labeled with ^99m^Tc, PSMA-targeted huJ591 DOTA (1,4,7,10-tetraazacyclodode cane-N,N,N,N-tetraacetic acid)-cysteine-modified scFv diabody labeled with ^64^Cu) and so far their usefulness was evaluated for prostate cancer diagnosis by PET and SPECT [84,88,89].

Monovalent single-chain variable region fragments (~25–30 kDa) represent the smallest antibody fragments, which are generated with VL and VH connected by a “flexible” linker. Although monovalent scFv fragments exhibit efficient tumor penetration, they are cleared rapidly from blood and can demonstrate poor antigen binding [38]. Recently, a comprehensive overview of the current literature on PSMA-targeted monovalent scFv fragments for radionuclide PET and SPECT imaging was presented by Diao et al. [12]. The highly promising preclinical results in vitro were reported for scFv constructs developed from the D2B antibody and radiolabeled with ^131^I (^131^I-scFvD2B), ^111^In (^111^In-scFvD2B), ^123^I (^123^I-scFvD2B) and ^124^I (^124^I-scFvD2B) [90,91,92,93]. The PSMA-targeted scFv constructs were also developed from the huJ591 antibody (DOTA-cysteine-modified scFv fragment labeled with ^64^Cu and its conjugate to DSPE-PEG micelles) for PET imaging [89]. Another studies reported that the scFvhuJ549 fragments labeled with ^68^Ga (^68^Ga-THPscFv), ^99m^Tc (^99m^Tc-scFv) and ^89^Zr (^89^Zr-scFv) showed also selective binding to the PSMA-expressing cells in vitro [84,94,95].

Nanobodies (~12–15 kDa) are antigen-binding fragments from heavy chain-only antibodies (VHH), derived from camelidae family [96]. They are the smallest possible functional antibody derivatives that combine high affinity with improved diffusion in tumor tissues, better pharmacokinetics, and a lower immunogenicity. One of the unique characteristics of nanobodies is their ability to target antigenic epitopes at locations, which are difficult to access by large molecules, such as conventional monoclonal antibodies [97]. So far, PSMA-targeted nanobodies were developed mainly for prostate cancer imaging and were tested in preclinical studies in vitro and in mice bearing xenograft. Zare et al. [98] designed the VHH nanobody against PSMA with a high in vitro specificity and affinity toward the PSMA on LNCaP cells. PSMA nanobodies labeled with ^99m^Tc (^99^mTc-labeled PSMA6 and PSMA30) and ^111^In for SPECT imaging, showing specific recognition of cell-expressed PSMA, good tumor penetration capability and fast clearance in PSMA-expressing xenografted mice [99]. Chatalic et al. [100] reported the ^111^In-DTPA-labeled engineered expression-modified nanobody JVZ-007, with a myc-tag and cys tag (^111^In-JVZ007-c-myc-his and ^111^InJVZ007-cys), which displayed good PSMA-positive tumor-targeting capability with rapid blood clearance in vivo. The therapeutic usefulness of the PSMA nanobody was evaluated for the first time in 2017 [101]. This study revealed an efficient and rapid prostate tumor-targeting by the JVZ-008 nanobody labeled with ^213^Bi (^213^Bi-JVZ-008) in mice bearing PSMA-positive LNCaP xenograft.

Monobodies (~10 kDa) are synthetic binding protein platforms based on the fibronectin type III (FN3) domain that has an immunoglobulin fold, but no disulfide bonds [102]. FN3 has a well-defined structure in which three solvent-accessible loops (BC, DE, and FG) are responsible for binding [103]. Monobodies are usually generated by introducing multiple mutations in a protein scaffold. Directed evolution approaches using molecular display technologies, enables one to efficiently generate a vast ensemble (“library”) of mutants and identify clones that bind to the target molecule of interest with high affinity and specificity [104]. Monobodies advantages for human trials such as a small size for tissue penetration, molecular stability with high melting temperatures (82 °C) and efficient bacterial production and an expected low immunity as a protein of human origin [105]. To date, only two monobodies (E1 and E10) that specifically bind EphA2 have been developed and tested for prostate cancer cells. In vivo optical imaging showed strong targeting of Cy5.5-labeled E1 to mouse tumor tissue induced by PC3 human prostate cancer cell line that expresses a high level of hEphA2 [106].

### 3.3. Small-Molecule PSMA Inhibitors

An interesting alternative for using antibodies and antibody-based molecules against the enzymatic domain of PSMA is development of small-molecule PSMA inhibitors (Figure 3).

Small-molecule PSMA inhibitors are zinc binding compounds attached to glutamate or glutamate isostere and fall into three families: (1) phosphonate-, phosphate-, and phosphoramidate compounds; (2) thiols; and (3) ureas [107]. Initial phosphonate, phosphate and phosphoramidate inhibitors were prepared at ZENECA and later at Guilford Pharmaceuticals [108,109]; however, they failed in early clinical translation. Urea-based compounds designed to inhibit PSMA in the brain were first reported by Kozikowski et al. [110]. Currently, the most advanced category are urea-based PSMA ligands, which usually consist of 3 components: a binding motif (glutamate-urea-lysine [Glu-urea-Lys]), a linker and a radiolabel-bearing moiety (chelator molecule for radiolabeling or a prosthetic group for fluorinated [22], iodinated [111] and astatinated [112] agents. The first small-molecule inhibitors of PSMA, based on the Glu-urea-Lys motif (^123^I-MIP-1072 and ^123^I-MIP-1095), were introduced into the clinic in 2008 for prostate cancer imaging [113,114]. They showed an advantage to localize rapidly in tumor lesions, including soft tissue and bone metastases. This success led to initiation of several preclinical studies and clinical trials with different small-molecule PSMA inhibitor, such as MIP-1404, PSMA-11, 2-(3-{1-carboxy-5-[(6-fluoro-pyridine-3-carbonyl)-amino]-pentyl}-ureido)-pentanedioic acid (DCFPyL), PSMA-617, PSMA-1007, and others. A few recently published reviews provide a comprehensive overview of the current status of small-molecule PSMA inhibitor radiotracers for targeted imaging [20,67,115]. Promising results of the clinical investigation of small-molecule PSMA inhibitors for diagnosis have led to the evaluation of these ligands as potential PSMA-targeted radiotherapeutics. Moreover, the commercial availability of PSMA-617 has facilitated initiation of prostate cancer therapy with radionuclide-labeled PSMA-617 in different world regions (see Section 3).

## 4. Radionuclides Investigated for Use in Targeted Prostate Cancer Therapy

Targeted radionuclide therapy is a growing and favorable treatment option for cancer because of the advantage of delivering a highly concentrated absorbed dose to the targeted tumor, while sparing the surrounding normal tissues. In addition, the selective ability of radionuclide therapy is advantageous in the treatment of systemic malignancy, such as in bone metastases, where a whole-body irradiation using external beam radiotherapy is impossible. Since administration of radionuclides is minimally invasive and duration of treatment is shorter than chemotherapy, PSMA-targeting ligands labeled with radionuclides have been evaluated in several preclinical and clinical studies, showing very promising results. In general, there are three types of radiation that can be used for targeted radionuclide therapy: β^−^-particles, α-particles and Auger electrons (Figure 4).

Each radionuclide is characterized by its own chemistry, half-life, decay properties, tissue range, and availability, giving the opportunity to adapt these features to a particular type of cancer and needs of an individual patient [116]. Despite the promising preliminary results obtained in the clinical studies, the costs and limited availability of several the best candidate therapeutic radioisotopes for widespread medical applications have limited their clinical use. To overcome this problem, intense research is going on presently for new radioisotope production at reasonable cost.

### 4.1. β^−^-Emitting Radionuclides

β^−^-particles are electrons emitted from the nucleus of decaying radioactive atoms (one electron/decay) that have various energies (from zero up to a maximum) and thus, a distribution of ranges. After their emission, the daughter nucleus has one more proton and one less neutron. However, because of small mass of electron, the recoil energy of decaying atom is negligible [117]. Due to relatively long range, β^−^-particles have a low linear energy transfer (LET) (0.2 keV/μm); however they are able to damage DNA, producing repairable DNA lesions, such as single- or double-stranded DNA breaks, base chemical modifications, and protein crosslinks. However, the damage caused by direct ionization of a target might only be sublethal, if dosed insufficiently high. Thus, indirect effects caused by reactive oxygen species (ROS) also contribute to the eventual damage. β^−^-particles have a relatively long range in tissue (1–10 mm), causing cytotoxic damage in surrounding nontargeted cells, referred to as ‘crossfire effect’. The main advantage of crossfire effect is that it negates the necessity of the radiotherapeutic agent being present within each of the targeted cells. This might be useful for the treatment of heterogeneous, bulky tumors, but it has the disadvantage of damaging surrounding normal tissue. Out of various known β^−^-radionuclides, yttrium-90 [118,119], lutetium-177 [120,121] iodine-131 [122] and terbium-161 [123] have been investigated so far for targeted prostate cancer immunotherapy (Table 1).

### 4.2. α-Emitting Radionuclides

Targeted radiotherapy based on α particles is a promising alternative to that based on β^−^-particles, because the α particles deposit the whole of their energy within a few cell diameters (50–100 µm). The α particle, a ^4^He nucleus, is relatively heavier than other subatomic particles emitted from decaying radionuclides, thus it has much shorter range in a tissue. Therefore, targeted α-radiotherapy agents have great potential for application to small, disseminated tumors and micro-metastases, and for treatment of hematological malignancies consisting of individual, circulating neoplastic cells. Compared with β^−^-particles, *α* particles provide a very high relative biological effectiveness, killing more cells with less radioactivity. Their high effectiveness results from induction of lethal DNA double strand breaks. Cell survival studies have shown that in contrast to β^−^-radiation, α particle-killed cells independently of their oxygenation state, cell cycle position or fluency [124]. Due to these advantages, targeted α-particle therapy is the most rapidly developing field in nuclear medicine and radiopharmacy [125]. Unfortunately in the case of radionuclides such as ^225^Ac, ^227^Th and ^223^Ra the daughter products are also α-emitters or β-emitters, and these radionuclides not remain complexed to chelators since they represent elements with different chemistry. In addition, the high recoil energy released during α-particle decay is about 10,000 times greater than the energy of a chemical bond and may easy disrupt the linkage between the α-emitter and the biomolecule [126]. Release of daughter radionuclides and their redistribution to normal tissues have been reported for the ^225^Ac which decays to several daughter radionuclides, including ^213^Bi, which is also an α-emitter [127]. The liberation of the recoiled radionuclides allows them to freely migrate in the body, causing toxicity to healthy tissues and decreasing the therapeutic dose delivered to the tumor. The renal toxicity induced by longer-lived decay product ^213^Bi is considered to be the major constraint to apply ^225^Ac in tumor therapy [128,129]. A review publication broadly describing recoil problem has been recently published by Kozempel et al. [125]. Several α emitters have been investigated so far for targeted prostate cancer immunotherapy: bismuth-213 [130,131], actinium-225 [125,132], astatine-211 [133], radium-223 [134,135], thorium-227 [136] and lead-212 [137] (Table 1). Among them, radium radionuclides have not yet found application in receptor-targeted therapy because of the lack of appropriate bifunctional ligands. Radium is a member of the 2 group of Periodic Table and similarly to other elements in this group does not form stable complexes. So far, several chelating agents have been evaluated for its complexation; however, the results were unsatisfactory [138]. Attempts have been made to incorporate ^223^Ra into liposomes but their application as carriers was not brought into practice because of low stability, relatively large diameters and necessity of labeling before conjugation with biomolecule [139]. Recently, the satisfactory immobilization of ^223^Ra in NaA nanozeolites [140], magnetite nanoparticles [141], polyoxopalladate [142], hydroxyapatites [143] and CaCO_3_ microparticles [144] has been developed.

### 4.3. Auger Electron Emitters

Auger electrons are extremely low-energy electrons with subcellular ranges (nanometers) emitted by radionuclides that decay by electron capture and/or internal conversion. The burst of low-energy electrons results in highly localized energy deposition (10^6^−10^9^ cGy) in an extremely small volume (several cubic nanometers) around the decay site and molecules in the immediate vicinity of the decaying atoms are irradiated by these electrons [145]. However, radionuclides that emit Auger electrons also release γ-rays, X-rays, β^−^-particles and internal conversion (IC) electrons. Hence, due to the diverse radiations, energy deposition distances, and the dimension of critical targets, which range from single cells and subcellular compartments, to tumor masses and normal organs, the whole picture of dosimetry for AE-emitting radionuclides is complicated. Moreover, an interesting phenomenon in the case of treatment by radionuclides that emit Auger electrons and α particles is the so-called bystander effect. It was found in cells which have not been directly irradiated following the irradiation of other nearby cells. A few mechanisms were proposed—one is the transfer of genomic instability through p53-mediated pathways and the other suggests that irradiated cells secrete cytokines or other factors that transit to other cells that were not irradiated and signals increased levels of intracellular reactive oxygen species [146]. So far, only one Auger electron-emitting radionuclide has been investigated for targeted prostate cancer immunotherapy, Iodine-125 [111] (Table 1). In addition, described above ^161^Tb radionuclide, which also emits small number of Auger electrons was used for labeling PSMA-617.

## 5. PSMA-Targeting Ligands Labeled with Radionuclides for Targeted Prostate Cancer Therapy

### 5.1. PSMA-Targeting Antibody-Based Molecules Conjugated to β^−^ Emitters

The earliest PSMA-targeted radioimmunoconjugate designed for prostate cancer patients was based on the 7E11 murine mAb (7E11/CYT-356, capromab pentide) labeled with the γ-emitter indium-111 (^111^In) [147]. This radioimmunoconjugate was approved by the Food and Drug Administration (Prostascint) for clinical detection of recurrent and metastatic prostate cancer in soft tissue. The next radioimmunoconjugate designed for targeted cancer prostate therapy was the anti-PSMA antibody CYT-356 (7E1 l-C5.3) radiolabeled with ^90^Y [148]. A Phase I dose-escalation study using ^90^Y-CYT-356 monoclonal antibody was performed in 12 patients with hormone-refractory prostate carcinoma. The dose of ^90^Y-CYT-356 was escalated by 3 mCi/m^2^ in group of three patients to determine a maximum tolerated dose. In addition, bio-distribution and immunogenicity of ^90^Y-CYT-356 mAb were studied. Unfortunately, clinical application of the ^90^Y-CYT-356 was unsuccessful due to a lack of therapeutic efficacy and presence of significant myelosuppression. Recognition of these features led to extensive preclinical studies of other murine antibodies (muJ591, muJ415, muJ533 and muE99) and humanized form of muJ591 (huJ591), which target the external domain of PSMA [149,150,151,152,153,154] (Table 2). These studies showed that murine mAbs lead the development of a human anti-mouse antibody (HAMA) response that precluded repetitive dosing. Therefore, mAb muJ591, showing the best specificity, was de-immunized into a humanized form (huJ591) and conjugated to ^111^In, ^131^I, ^90^Y, and ^177^Lu. Among them, ^90^Y and ^177^Lu labeled mAbs provided better dosimetry than ^111^In and ^131^I due to their longer intracellular half-lives and better clearance. Anti-tumor responses were seen with these radionuclides with an apparent dose–response relationship. Higher cumulative doses of either ^90^Y or ^177^Lu could be delivered using fractionated, sub-maximum tolerated doses rather than a single maximum tolerated. Moreover, median survival of the animals improved significantly for fractionated therapy. Based on the promising results from preclinical studies, several groups started treatment of patients suffering from mCRPC with PSMA-targeting antibody-based molecules conjugated to β-emitters [155,156,157,158,159,160,161,162,163,164,165,166] (Table 3).

In general, the clinical experience from these studies have shown a significant dose-dependent treatment response with relatively low level of toxicity. However, despite the fact that over the last 20 years great progress has been made in the field of PSMA-targeted radionuclide prostate cancer therapy, there is still only a small number of clinical trials including approximately 400 mCRCP patients treated with antibody-based molecules conjugated to β^−^ emitters. Moreover, most of these studies were classic dose-escalation phase I studies. All currently published studies with ^90^Y-huJ591 and ^177^Lu-huJ591 therapy involve a variety of treatment regimens, both in terms of dose given (ranging from 0.185–0.740 GBq/m^2^ of ^90^Y-huJ591 and from 0.370 to 3.3 GBq/m^2^ of ^177^Lu-huJ591) and the number of doses administered, including single injections, escalation doses, and fractionated doses. This makes interpretation of the efficacy of the treatment difficult at this stage. In the currently published literature, the number of men who experienced >50% reduction in serum PSA levels ranges from 30% to 70%. Number of patients, who did not respond to ^90^Y-huJ591 and ^177^Lu-huJ591 therapy ranges from 10% to 32%. This is likely due to a variety of factors. One of the most important factors is probably a heterogenous density of the PSMA receptor among patients. Due to the limited number of patients, there is also little information on a possible survival benefit of these therapies. Available data revealed that median progression-free survival was 13.7 months. However, the most commonly reported serious side effect related to ^90^Y-huJ591 and ^177^Lu-huJ591 therapies is hematological toxicity. Reversible myelosuppression, including thrombocytopenia and neutropenia has been reported in many patients. The cause of hematological toxicity is probably poor penetration of large antibodies into solid tumors and their slow clearance from the circulation. None of the current studies have reported renal toxicity, although it is supposed, that this may be a long-term complication. Kidney absorbed dose related to ^90^Y-huJ591 and ^177^Lu-huJ591 therapy has generally been reported as being well below dose-limiting threshold. A greater number of clinical trials should allow better understanding of potential benefits and harms of targeted therapies of mCRPC using antibody-based molecules conjugated to β^−^ emitters. Currently a randomized phase II multi-center trial, designed to evaluate the clinical utility of ^177^Lu-J591 in combination with ketoconazole in patients with high risk castrate biochemically relapsed PCa after local therapy [NCT00859781] is in progress (Table 4)

### 5.2. PSMA-Targeting Small Molecules Conjugated to β^−^ Emitters

In 2009, Maresca and co-workers described the design and synthesis of halogenated heterodimeric inhibitors of prostate-specific membrane antigen (PSMA) for targeting prostate cancer [114]. On the basis of this work, Hillier et al. (2009) evaluated two of the most potent radioiodinated compounds, ^123^I-MIP-1072 ((*S*)-2-(3-((*S*)-1-carboxy-5-(4–iodobenzylamino) pentyl)ureido) pentane dioic acid) and ^123^I-MIP-1095 ((*S*)-2-(3-((*S*)-1-carboxy-5-(3-(4-iodophenyl) ureido) pentyl) ureido)pentanedioic acid), for their ability to bind to PSMA and localize to PSMA-expressing tumors in vivo [113]. These ligands potently inhibited the glutamate carboxypeptidase activity of PSMA and when radiolabeled with ^123^I exhibited high affinity for PSMA on human prostate cancer LNCaP cells. Tissue distribution studies in mice demonstrated that ^123^I-MIP-1095 exhibited greater tumor uptake but slower washout from blood and non-target tissues compared to [^123^I]MIP-1072. In 2014 the research group at Technical University Munich developed the ^177^Lu-PSMA I&T, which may be radiolabeled with a variety of diagnostic and therapeutic radiometals [167,168]. ^177^Lu-PSMA I&T showed high PSMA-affinity and efficient internalization into PSMA-expressing cells and tumor lesions. In 2015 Heidelberg group developed ^177^Lu-PSMA-617, which exhibited an excellent cancer prostate cells and tumor-targeting as well as pharmacokinetic properties [115]. One of the earliest patients treated with two cycles of ^177^Lu-PSMA-617 with a cumulative activity of 7.4 GBq presented a radiological complete response [169]. Based on the preclinical and first in-human studies on PSMA-targeting small molecules conjugated to β^−^-emitters, several clinical trials have been performed [170,171,172,173,174,175,176,177,178,179,180,181,182,183,184,185] (Table 3).

In general, the clinical experience from these studies have shown that these therapies can be an effective treatment modality when all other available treatment options have failed encouraging efficacy. To date more than 400 patients have received ^177^Lu-PSMA-617 or ^177^Lu-PSMA I&T radioconjugates for targeted therapy of mCRPC. However, the reported clinical trials present remarkable heterogeneity, including previous therapies of patients, stage of disease, site of metastasis and a variety of treatment regimens, both in terms of dose given (ranging from 1.1 to 14.8 GBq for ^177^Lu-PSMA-617 and from 2 to 9.7 GBq for ^177^Lu-PSMA I&T) and the number of doses administered, including single injections, escalation doses, and fractionated doses. This makes interpretation of the efficacy of the treatment difficult at this stage. Using PSA as a response parameter, these studies have revealed that among all treated patients, approximately 80% of them showed PSA response in terms of any PSA decline, whereas approximately 60% presented with a PSA decline of ≥ 50%. Patients, who did not respond to ^177^Lu-PSMA-617 and ^177^Lu-PSMA I&T therapies range from 20% to 36%. Available data have revealed notable differences in patient’s median progression-free survival (from 6.3 to 12 months) and median overall survival (from 12.7 to 60 weeks). No nephrotoxicity or hepatotoxicity has been observed and only very few cases of grade 3–4 hematological toxicity. Hematological toxicity included transient and self-limiting declines in hemoglobin, neutrophils, and platelets. No grade 3/4 non-hematological toxicities were observed. Grade 1/2 non-hematological events included e.g., xerostomia, nausea, vomiting, and bone pain. Overall, these results demonstrate that ^177^Lu-PSMA-617 and ^177^Lu-PSMA I&T therapies represent a promising new therapeutic option for mCRPC. To better understand the differences in bio-distribution, therapeutic efficacy and toxicity of ^177^Lu-PSMA-617 and ^177^Lu-PSMA I&T therapies, several randomized studies on progressive PSMA-positive mCRPC treated with these therapies 617 are in progress (Table 4).

### 5.3. PSMA-Targeted Antibody-Based Molecules Conjugated to α-Emitters

Despite the fact that several clinical studies have shown encouraging results of treatment with PSMA ligands labeled with the beta emitter ^177^Lu in patients with advanced mCRPC, it was found that up to 30% of patients never responded to the therapy or developed resistance [186]. Because of this and a theoretical physical and biological advantage of α- versus β-particle emitters, various centers constructed and tested the PSMA-targeted ligands conjugated to α-particle radionuclides.

McDevitt et al. [187] described for the first time the construction of the α-particle-emitting agent ^213^Bi-J591 for targeted prostate cancer therapy. In vitro results revealed efficient binding and internationalization of ^213^Bi-J591 into LNCaP cells. However, in vivo experiments showed that a single course of the ^213^Bi-J591 improved significantly median tumor-free survival and reduced PSA level in athymic nude mice bearing prostate cancer xenografts. These findings have subsequently prompted further preclinical in vitro and in vivo research with others α-particle-emitting radionuclides [187,188,189,190,191] (Table 2).

To our knowledge, no clinical studies have been performed so far to investigate the therapeutic efficiency and safety of PSMA-targeted antibody-based molecules conjugated to α-emitters. However, a phase I dose−escalation trial of ^225^Ac−J591 in 42 Patients With Metastatic Castration−Resistant Prostate Cancer [NCT03276572] is currently underway with estimated study completion date at July 2021 (Table 4).

### 5.4. PSMA-Targeted Small Molecules Conjugated to α-Emitters

In 2013, the first ^225^Ac-labeled PSMA small-molecule ligand PSMA-617 for targeted prostate cancer therapy was developed and investigated at Joint Research Center (JRC) Karlsruhe. First in-human study indicated remarkable response in 2 patients with mCRPC patients that were progressive under ^177^Lu-PSMA-617 therapy [192]. Patient A was treated with 3 cycles of 9–10 MBq (100 kBq per kg b.w.) of ^225^Ac-PSMA-617 at bimonthly intervals. Post therapeutic emission scans validated sufficient tumor-targeting. Two months later, all previously PSMA-positive lesions had visually disappeared and accordingly, the PSA level had dropped from more than 3000 ng/mL to 0.26 ng/mL. The patient received an additional 6 MBq of ^225^Ac-PSMA-617, resulting in a further PSA decline to less than 0.1 ng/mL along with a complete imaging response. Patient B received 3 cycles of 6.4 MBq (100 kBq per kg b.w.) at bimonthly intervals. Restaging based on the PSMA PET/CT results finally indicated a partial response after 2 cycles and a complete remission after 3 cycles with no relevant hematologic toxicity. However, severe xerostomia occurred and became the dose-limiting toxicity.

In a second first in-human ^225^Ac-PSMA-617 study, 14 patients received more than one cycle of treatment using 50, 100, 150 and 200 kBq/kg b.w. doses at intervals of 2–4 months [193]. All patients in the 200 kBq/kg b.w. group and 1 of 2 patients in the 150 kBq/kg group discontinued therapy due to intolerance of the treatment. The authors considered 100 kBq/kg b.w. as a maximal tolerable dose, with xerostomia being dose-limiting factor. The decline of PSA > 50% was seen in 4 of 9 patients with no renal or liver toxicity. Overall survival was a median of 8 months in this group of patients. Following these preliminary studies of ^225^Ac-PSMA-617 at JRC Karlsruhe, clinical trials were started initially in collaboration with University Hospital Heidelberg, and subsequently extended to Technical University Munich and Steve Biko Academic Hospital Pretoria [194,195,196,197] (Table 3).

To date, anti-tumor activity of ^225^Ac-PSMA-617 was demonstrated in 4 clinical trials, including 150 patients with mCRPC. As far as a reliable interpretation can be allowed (considering the small number of patients and differences in their characteristics), these clinical data have shown that therapy with ^225^Ac-PSMA-617 can overcome resistance to therapy with beta emitters and can offer a valuable additional treatment option to patients that have failed established treatment. These results have also revealed that among all treated patients, almost 90% of them showed PSA response in terms of any PSA decline, whereas approximately 65% presented with a PSA decline of ≥50%. In one study, 82% presented with a decline of >90%. Important advantages of using ^225^Ac-PSMA-617 included fast tumor uptake and rapid clearance of unbound conjugates from circulation, leading to the reduced hematological toxicity and toxicity to the kidneys. Moreover, because of the short tissue-penetration range of 225Ac-PSMA-617, it had an advantage regarding very low hematologic toxicity in patients with diffuse-type bone marrow infiltration. One critical observation during these studies was the high number of patients who, despite a promising PSA response, discontinued therapy because of intolerable xerostomia. Clinical trials with larger patient populations are needed to further investigate the efficacy, safety, and patient selection for ^225^Ac-PSMA-617.

In 2017, Sathekge et al. [198] presented the first in-human treatment concept with ^213^Bi-PSMA-617 in a single patient from South Africa with mCRPC that was progressive under conventional therapy. The patient was treated with two cycles of ^213^Bi-PSMA-617 with a cumulative activity of 592 MBq. Restaging with ^68^Ga-PSMA PET/CT after 11 months showed a remarkable molecular imaging response. This patient also demonstrated a significant decrease in PSA level. Not significant toxicities were reported, and no cross-sectional imaging was reported. Dosimetry estimates have been reported with ^213^Bi-PSMA-617 in three patients [199]. A relative biological effectiveness of 5 for the 8.4 MeV alpha ^213^Bi emission was used in the dosimetry calculations. The authors concluded that ^213^Bi, as compared to ^225^Ac, is a “second choice” when using PSMA-617 as a ligand.

As mentioned before ^225^Ac availability is limited to several clinical trials per year. However, promising results obtained in clinical trials with ^225^Ac-PSMA targeted conjugates directed researchers to search for other more readily available α-emitters. One of such radionuclides is commercially available ^212^Pb. For therapeutic application Banerjee and co-workers [144] designed four conjugates containing DOTA chelator connected to low-molecular-weight ligands synthesized using the lysine-urea-glutamate scaffold. It was found using γ-emitting ^203^Pb that PSMA tumor lesions were visualized by SPECT/CT, as early as 0.5 h after injection. Preclinical study using micrometastatic tumor models with a single administration of ^212^Pb demonstrated increased survival benefit compared to ^177^Lu-PSMA-617. The kidney was identified as the dose-limiting organ from the long-term toxicity study.

### 5.5. PSMA-Targeting Ligands Conjugated to Auger Electron Emitters

To date, only two PSMA-targeting ligands conjugated to Auger electron emitters have been developed and tested in preclinical studies [107,123] (Table 2). In 2015, Kiess et al. [107] developed and investigated a highly specific small-molecule Auger emitter targeting PSMA, 2-[3-[1-carboxy-5-(4-^125^I-iodo-benzoylamino)-pentyl]-ureido]-pentanedioic acid (^125^I-DCIBzL). This study showed the PSMA-specific cellular uptake and cytotoxicity of this conjugate in vitro and efficiency in induction of DNA damage and tumor growth delay in mice bearing PSMA+ PC3 PIP tumor xenografts.

More recently Müller et al. [123] investigated ^161^Tb in combination with PSMA-617, as a potentially more effective therapeutic alternative to ^177^Lu-PSMA-617. The in vitro findings revealed enhanced therapeutic effects of ^161^Tb compared to ^177^Lu. In agreement with dosimetric calculations, ^161^Tb-PSMA-617 was up to 3-fold more effective than ^177^Lu-PSMA-617 in vitro. It was also confirmed that pharmacokinetics of ^161^Tb-PSMA-617 was equal to ^177^Lu-PSMA617, resulting in the same bio-distribution profiles. The treatment of mice with 5 MBq or 10 MBq ^161^Tb-PSMA-617, showed an activity-dependent tumor growth inhibition and prolonged The results revealed the improved effect of ^161^Tb over ^177^Lu at the level of single cancer cells and cancer cell clusters in vivo. Based on these promising preclinical data, the same group demonstrated the successful preparation and preclinical testing of ^152^Tb-PSMA-617 as well as its first application for PET/CT imaging of prostate cancer [200].

## 6. PSMA-Targeted Nanoparticle-Based Radiopharmaceuticals

Nanoparticles represent an emerging technology in medicinal applications due their unique pharmacokinetic properties, amenability for multifunctionalization and high loading capacities [201]. With respect to PSMA-targeting, several types of nanoparticles have been outfitted with various types targeting agents (e.g., antibodies, aptamers, urea inhibitors) demonstrating their utility for in vitro and in vivo prostate cancer applications [202,203,204]. However, even though nanoparticles seem to be attractive platform for the development of multimodal theragnostic agents, not many nanoparticle-based radiopharmaceuticals for prostate cancer therapy have been reported.

In a study by Bandekar et al. [205] the potential of targeted liposomes loaded with the alpha particle generator actinium-225 was evaluated to selectively kill PSMA-expressing cells in vitro. These nano-sized carriers (~107 nm) were composed of 1,2-dinonadecanoyl-sn-glycero-3-phosphocholine(DNPC):cholesterol:1,2-distearoyl-sn-glycero-3-phosphoethanolamine-*N*-[methoxy-polyethyleneglycol-2000](DSPE-PEG):1,2-dipalmitoyl-sn-glycero-3-phosphoethanolamine-*N*-(lissamine rhodamine B sulfonyl) (DPPE-rhodamine) and labeled with two different types of PSMA-targeting ligands: the anti-PSMA antibody J591 and the A10 PSMA aptamer, both recognizing the extracellular domain of the PSMA protein. The conjugation of PSMA-targeting ligands resulted in a density of approximately 17 J591 antibodies or 9 A10 aptamers per liposome. The authors stably encapsulated up to three ^225^Ac nuclides per every 2 liposomes and were able to prepare as high as 1.15 kBq (31 nCi) per 0.3 mg of antibody, corresponding to one ^225^Ac per 840 antibodies. The targeting selectivity, extent of internalization, and killing efficacy of liposomes were evaluated in prostate cancer cells intrinsically expressing PSMA (human LNCaP and rat Mat-Lu cells) and in human umbilical vein endothelial cells induced to express PSMA (induced HUVEC), exposed to ^225^Ac-loaded liposomes (activity from 37 × 10^−9^ to 370 kBq/mL (10^−9^ to 10^1^ mCi/mL). These studies demonstrated the superiority of J591-labeled liposomes over both A10 PSMA-labeled liposomes and nontargeted liposomes in terms of their binding and internalization efficacies as well as the observed radioactivity and cytotoxicity.

Zhu et al. [206] designed lipid-based nanocarriers composed of 21PC (2-diheneicosanoyl-sn-glycero-3-phosphocholine): Cholesterol: DSPE-PEG(1,2-distearoyl-sn-glycero-3-phosphoethanolami ne-*N*-[Methoxy (Polyethylene glycol)-2000]:DPPE-Rhodamine (1,2)-dipalmi-toyl-sn-glycero-3-phosphoethanolamine-*N*-(LissamineRhodamine B Sulfonyl). These nanocarriers were loaded with actinium-225 and labeled with two different types of PSMA-targeting ligands: a fully human PSMA antibody (mAb) and a urea-based low-molecular-weight agent, both targeting the PSMA on human endothelial cells. The average size of these nanocarriers was 107 nm with a density of approximately 30 antibodies or 368 urea-based ligands per nanocarrier. Cell association of delivered radioactivity and cell viability studies were performed in LNCaP, PSMA-HUVEC and PSMA + HUVEC cells exposed to ^225^Ac-loaded liposomes at a lipid concentration of 83.33 μM and activity of 37 kBq/mL. The authors reported that both types of nanocarriers improved the killing efficacy of delivered activity per cell by almost three fold relative to the killing efficacy of the same levels of activity when delivered by the PSMA-targeting antibody without nanocarrier. The increase in killing efficacy, which was accompanied by increased levels of DNA double strand breaks, strongly correlated with intracellular patterns of nanocarriers exhibiting nucleo-cytoplasmic localization, unlike the antibody without nanocarrier, which preferentially localized near the plasma membrane.

## 7. Conclusions

Although the scientific rationale for the PSMA-targeted radionuclide therapies in men with mCRPC appears to be well supported, numerous questions remain to be answered before extensive application of them in routine clinical practice. Among them, implication of PSMA heterogeneity for treatment responses and selection and/or improving already existing PSMA-targeted molecules are still a matter of debate. The most commonly used J591 antibody presents high tumor and low healthy organs uptake, but suffers from slow accumulation kinetics. On the contrary, the most commonly used PSMA-617 and PSMA I&T small-molecule inhibitors possess high specificity and fast pharmacokinetics, but suffer from accumulation in healthy organs, causing radiotoxicity. An important issue is also identification of optimal radionuclide and the appropriate administered activity for the PSMA-targeted radionuclide therapies. Recent studies proved the increased therapeutic effectiveness of alpha radiation compared to beta radiation, but alpha radiation therapy is accompanied by an increased risk of side effects. Another open question to be taken into account is standardization of dosimetry protocols to assess safety and efficacy of the PSMA-targeted radionuclide therapies. In recognition of all the pros and cons of the PSMA-targeted radionuclide therapies, The Prostate Cancer Foundation convened in 2017 a PSMA-Directed Radionuclide Scientific Working Group to discuss the potential for using PSMA-targeted radionuclide agents for the treatment of advanced prostate cancer and to define the future studies and clinical trials necessary for validating and optimizing the use of these. The Working Group Members from several academic institutions in the USA, Australia, and Europe and from the National Institutes of Health/National Cancer Institute (USA) and Bayer Pharmaceuticals highlighted barriers to successful use these agents in different practice models agents [207]. It can be expected that the lessons learned from this Working Group meeting as well as from ongoing and future clinical trials will be translated into clinical practice to improve patient care and treatment.

## Figures and Tables

**Figure 1 molecules-25-01743-f001:**
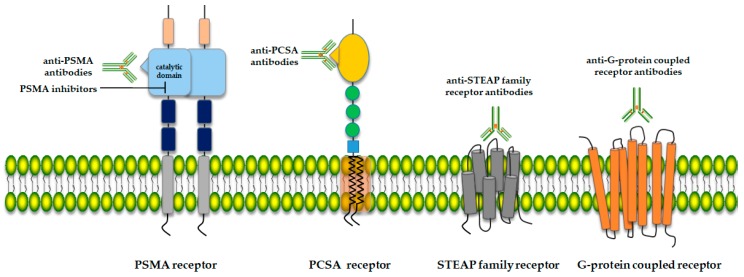
Schematic structure of selected prostate cell membrane receptors as potential prostate cancer therapy targets.

**Figure 2 molecules-25-01743-f002:**
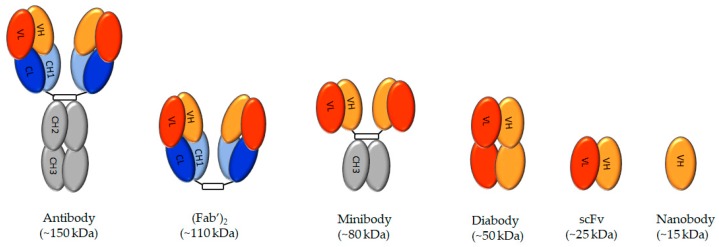
Schematic representation of antibody-based molecules. VH—heavy chain variable region, VL—light-chain variable region, CL—light-chain constant region, CH—high chain constant region 1, CH2—high chain constant region 2. CH3—high chain constant region 3.

**Figure 3 molecules-25-01743-f003:**
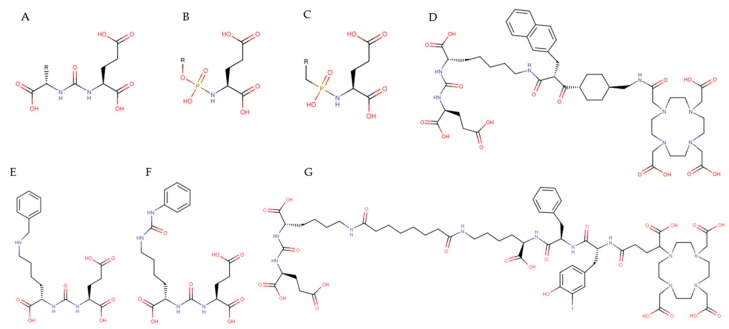
Chemical structure of selected PSMA-targeted small-molecule inhibitors. (**A**) urea-based compounds, (**B**) glutamatephosphoramidates, (**C**) 2-(phosphinylmethyl)pentanedioic acids, (**D**) PSMA-617, (**E**) MIP-1072, (**F**) MIP-1095, (**G**) PSMA I&T.

**Figure 4 molecules-25-01743-f004:**
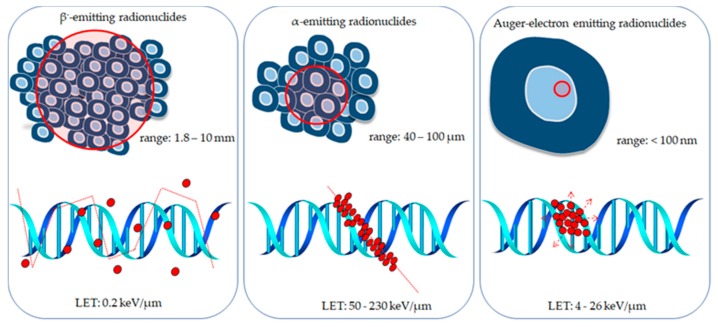
Schematic representation of the tissue-penetration range and density of ionization events caused by β^−^-particles, α-particles, and Auger electrons.

**Table 1 molecules-25-01743-t001:** Beta^−^-particle, alpha-particle, and Auger electron-emitting radionuclides for targeted prostate cancer therapy.

Radionuclide	Emitted Particle	Half-Life	Maximum Range in Tissue [mm]	Energy of Emitted Particle [MeV]	Production Mode	Availability	References
Yttrium-90	β^−^	2.67 days	12	2.28	^90^Sr/^90^Y generator	Commercially available (low price)	[118,119]
Lutetium-177	β^−^	6,65 days	1.6	0.49	^176^Lu(n,γ)^177^Lu	Commercially available (low price)	[120,121]
					^176^Yb(n,γ) ^177^Yb $\overset{\beta^{-}}{\to }$^177^Lu	Commercially available (high price)	
Iodine-131	β^−^/γ	8.02 days	2.3	0.97	^nat^Te(n,γ)^131^I	Commercially available (low price)	[122]
Terbium-161	β^−^/Auger and CE	6.89 days	0.03	0.15	^160^Gd(n,γ)^161^Gd $\overset{\beta^{-}}{\to }$^161^Tb	Low availability, difficult production	[123]
Bismuth-213	α/β^−^	45.6 min.	0.084	8.38	^225^Ac/^213^Bi generator	Moderate availability	[130,131]
Actinium-225	α	10.0 days	0.061	28	^229^Th/^225^Ac ^226^Ra(p,2n)^225^Ac ^232^Th(p,spall.) ^225^Ac	Moderate availability	[125,132]
Astatine-211	α	7.2 days	0.067	5.87	^nat^Bi(α,2n)^211^At	Moderate availability	[133]
Radium-223	α	11.43 days	0.08	28.2	^227^Ac/^223^Ra generator	Commercially available	[134,135]
Thorium-227	α	18.7 days	0.1	6.14	^227^Ac/^227^Th generator	Moderate availability	[136]
Lead-212	β^−^/αdecays to ^212^Bi	10.6 h	0.08	6.05	^228^Th/^224^Ra/^212^Pb generator.	Commercially available	[137]
Iodine-125	Auger and CE	59.4 days	0.0001	0.035	^124^Xe(n,γ)^125^Xe $\overset{EC}{\to }$^125^I.	Commercially available (moderate price)	[111]

**Table 2 molecules-25-01743-t002:** Preclinical in vitro and in vivo studies of radiopharmaceuticals for targeted prostate cancer therapy.

Agent	Administered Activity	Animals Cell Type	Main Findings	Ref.
^131^I-J415, ^131^I-J533 ^131^I-J591, ^111^In-J415 ^111^In-J533,^111^In-J591	350 MBq/mg	LNCaP cells	J415 and J591 mAbs competed for binding to PSMA antigen, J533 did not interfered with J415, rapid elimination of ^131^I from the cell and high retention of ^111^In	[149]
^177^Lu-huJ591 mAb	10 µCi 100–400 µCi	BALB/c nude mice bearing LNCaP tumor xenografts	high uptake of ^177^Lu-huJ591 in tumors, lack of bone radioactivity, high stability of conjugate, dose-dependent tumor remission	[150]
^131^I-J415, ^131^I-J533 ^131^I-J591, ^131^I-1I-7E11 ^111^In-J415, ^111^In-J533, ^111^In-J591, ^111^In 1I-7E11	80 kBq	BALB/c nude mice bearing LNCaP tumor xenografts	^131^I-J533 showed lower tumor localization and reduced tumor/blood and tumor/muscle ratios than ^131^I-J415 or ^131^I-J591, better tumor uptake of ^111^In-labeled mAbs compared to ^131^I labeled mAbs	[151]
^131^I-huJ591 mAb ^90^Y-huJ591 mAb	3.7–11.1 MBq ^131^I-huJ591; 1.11–7.4 MBq ^90^Y-huJ591	BALB/c nude mice bearing LNCaP tumor xenografts	anti-tumor effects dependent on a size of tumor, radionuclide used and dose, median survival time increased, total dose, and dose rate equally important for bone marrow toxicity	[152]
^111^In-CHX-A-3C6 mAb	7.5 μCi 100 μCi	LNCaP, 22Rv, DU145 cells and BALB/c nude mice bearing LNCaP, 22Rv1, or DU145 tumor xenografts	high ability to bind to LNCaP and 22Rv1, but not to DU145 cells, excellent in vivo PSMA-targeting of ^111^In-CHX-A′′-3C6 in mice bearing LNCaP, 22Rv1 xenografts	[153]
^177^Lu-DOTA-3/F11	300 kBq0.5, 1, or 2 MBq	SCID mice bearing C4-2 tumor xenografts	increasing tumor uptake over time. Single dose of 1 MBq ^177^Lu-DOTA-3/F11 inhibited tumor growth and prolonged survival	[154]
^213^Bi-J591	0.06–6.4 mCi/mg	5E4 LNCaP cells, spheroids, nude mice bearing LNCaP tumors	efficient binding and internationalization of ^213^Bi-J591, improved median tumor-free survival, PSA decline	[187]
^213^Bi-J591	0.9 and 1.8 MBq/ml	LNCaP-LN3 spheroids	reduction in spheroid volume with increasing radioactivity	[188]
^213^Bi-J591	1–100 mCi	LNCaP-LN3 cells, nude mice bearing LNCaP tumors	in vitro very high cytotoxicity with increasing activity, inhibition of tumor growth, no side effects	[189]
^227^Th-PSMA IgG1	100–500 kBq/kg	nude mice bearing LNCaP-luc, C4-2 or 22Rv1tumors	selective anti-tumor efficacy at 250 and 500 kBq/kg, prevention of tumor growth at 100 kBq/kg	[190]
^227^Th-PSMA IgG1	75–500 kBq/kg	nude mice bearing PDx or KuCap1 tumors	dose-dependent tumor growth inhibition, stable disease or tumor regression at 300 kBq/kg, partial or complete regression of tumor growth at 250 or 500 kBq/kg in the PDx model	[191]
^211^At-6	3.7 kBq/100 μL 37–740 kBq	PC3 PIP and PC3-flu cells, nude mice bearing PC3 tumors	efficient cellular uptake in PC3 PIP tumors and kidneys, tumor growth delay in PC3 PIP xenograft model and improved survival in micrometastatic PC model	[112]
^213^Bi-PSMA I&T, ^213^Bi-JVZ-008	0.3 MBq5.4–6.6 MBq ^213^Bi-PSMA I&T 4.5–5.4 MBq ^213^Bi-JVZ-008	LNCaP cells, BALB/c nude mice bearing LNCaP tumors	induced DSBs in vitro, 2x higher tumor uptake of ^213^Bi-PSMA I&T than ^213^Bi-JVZ-008, ^213^Bi-PSMA I&T more potent to induce DNA damage in the tumor, possible nephrotoxicity	[101]
^125^I-DCIBzL	3.7–370 kBq/mL 3.7–111 MBq	PC3 cells, nude mice bearing PC3 tumors	increased DNA damage in vitro, significant tumor growth delay	[107]
^161^Tb-PSMA-617 ^177^Lu-PSMA-617	0.01–20 MBq/mL 5 or 10 MBq	PC3 cells, nude mice bearing PC3 tumors	enhanced therapeutic effects of ^161^Tb compared to ^177^Lu, equal pharmacokinetics of ^161^Tb-PSMA-617 and ^177^Lu-PSMA617	[123]

**Table 3 molecules-25-01743-t003:** Clinical studies of radiopharmaceuticals for targeted prostate cancer therapy (DLT—dose limiting toxicity; MTD—maximal toxic dose; PD—progressive disease, PFS—progression-free survival; MOS—median overall survival; RI—radioimmunotherapy).

Agent	Treatment Dose	Number of Patients	Main Findings	Ref.
^90^Y-huJ591	0.185–0.740 GBq/m^2^	29	DLT—0.74 MBq/m^2^, MTD—0.65 MBq/m^2^, myelosuppression was the DLT, excellent targeting of prostate cancer metastases	[155,156,157,158]
^177^Lu-huJ591	0.370 to 2.775 GBq/m^2^	24	myelosuppression was the DLT, excellent targeting of prostate cancer metastases	[159,160]
^177^Lu-huJ591	2.4 GBq/m^2^ 2.6 GBq/m^2^	47	dose–response relationship in PSA decline (71% vs. 46% of patients), MTD—2.59 MBq/m^2^, acceptable mielosuppresion, MOS—11.9 months with 2.4 MBq/m^2^ and 21.8 months with 2.6 MBq/m^2^	[161,162]
^177^Lu-huJ591	3.3 GBq/m^2^, fractionated doses: 0.74 GBq/m^2^ escalation dose: 0.18 GBq/m^2^	28	MTD—1.48 MBq/m^2^, fractionated dosing allowed higher cumulative doses of 177Lu-J591 with less toxicity	[163]
^90^Y-huJ591 ^177^Lu-huJ591	0.185–0.740 GBq/m^2^ 0.740–3.3 GBq/m^2^	121 29	myelosuppression was the DLT, patients could tolerate subsequent therapies after RIT	[164]
docetaxel/ ^177^Lu-J591	docetaxel (75mg/m^2^) with escalation doses of ^177^Lu-J591 (0.74–1.48 GBq/m^2^)	15	PSA decline in 80% (>50% in 73% of patients (and >50% in 73%), toxicity limited to reversible myelosuppression	[165]
^177^Lu-J591	0.74–1.67 GBq/m^2^	49	fractionated dosing allowed higher doses of 177Lu-J591 with less toxicity, myelosuppression was the DLT	[166]
^131^I-MIP-1095	8 GBq (range: 2–7.2 GBq)	28	PSA decline in 61% of patients, PD (14%), high doses received by the salivary glands, liver, and kidneys, reduction of pain	[170]
^177^Lu-PSMA I&T	7.4 GBq/cycle	18	the kidneys and glandular tissue were the critical organs, with a mean absorbed dose of 0.72 Gy/GBq and 3.8 Gy/GBq, respectively	[171]
^177^Lu-PSMA I&T	5.8 GBq/cycle	56	PSA decline in 80% of patients, PD (36%), no long-term side effects	[172]
^177^Lu-PSMA I&T	6.0 GBq/cycle (range: 2 to 9.7 GBq)	119	PSA decline in 76.3% of patients (>50% decline in PSA in 57.5%), no long-term side effects	[173]
^177^Lu-PSMA-617	5.6 GBq (range: 4.1–6.1 GBq)	10	PSA decline in 70% of patients, no long-term side effects	[174]
^177^Lu-PSMA-617	6 GBq/cycle	22	PSA decline in 80% of patients (>50% decline in PSA in 45%), PD (32%)	[175]
^177^Lu-PSMA-617	4.8 GBq/cycle	30	PSA decline in 70% of patients (>50% decline in PSA in 43.3%), PD (21%)	[176]
^177^Lu-PSMA-617	5.9 GB/cycles	82	PSA decline in 53% of patients, PD (21%), therapy was well tolerated	[177]
^177^Lu-PSMA-617	5.52 GBq (range: 5.28 to 5.77 GBq)	9	lacrimal glands represent the dose-limiting organs	[178]
^177^Lu-PSMA-617	3.6 GBq (range: 3.4 to 3.9 GBq)	5	salivary glands represent the dose-limiting organs	[179]
^177^Lu-PSMA-617	6.0 GBq	10	PSA decline in 47% of patients, PD (33%), no acute events for salivary gland or kidney,	[180]
^177^Lu-PSMA-617	1.11–5.55 GBq	31	PSA decline by 75% (mean for all patients), PD (20%), PFS—12 months, MOS—15 months	[181]
^177^Lu-PSMA-617	6.1 GBq (range: 5.4 to 6.5 GBq)	10	PSA decline in 50% patients, PD (30%), no side effects, xerostomia observed only in 1 patient	[182]
^177^Lu-PSMA-617	4–8 GBq	30	PSA decline in 97% patients (>50% decline in PSA in 57%) PFS—7.6 months, MOS—13.5 months	[183]
^177^Lu-PSMA I&T^177^Lu-PSMA-617	≥14.8 GBq	45	PFS better with higher dose, re-challenge therapy at progression decreased tumor grade, mild and transitory adverse effects	[184]
^177^Lu-PSMA-617	2–8 GBq	145	PSA decline > 50% in 45% patients, mild side effects	[185]
^225^Ac-PSMA-617	100 kBq/kg b.w.	40	PSA decline > 50% in 63% patients, xerostomia was the main side effect, PFS—7 months, MOS > 12 months	[194]
^225^Ac-PSMA-617^177^Lu-PSMA-617tandem therapy	5.3 MBq (range: 1.5 to 7.9 MBq) 6.9 GBq (range: 5.0–11.6 GBq)	20	PSA decline > 50% in 65% patients, PFS—19 weeks, OS—48 weeks tandem therapy enhanced response to therapy and decreased xerostomia severity	[195]
^225^Ac-PSMA-617	4–8 MBq	17	PSA decline in 94% of patients (>90% decline in PSA in 82%), grade 1/2 xerostomia in all patients	[196]
^225^Ac-PSMA-617	4–8 MBq	73	PSA decline in 83% (>50% decline in PSA in 70%), PD (32%), PFS—15.2 months, OS—18 months, xerostomia in 85% of patients	[197]

**Table 4 molecules-25-01743-t004:** Ongoing clinical trials of radiopharmaceuticals for targeted prostate cancer therapy.

Trial Identification	Agent	Phase	Study Arms	Estimated Enrolment	Primary Endpoints
NCT03828838	^177^Lu-PSMA-617	I/II	two cycles (3 GBq and 3–6 GBq)	10	doses delivered to the tumors and organs at risk
NCT 03511664	^177^Lu-PSMA-617	III	7.4 GBq (±10%) 177Lu-PSMA-617 every 6 weeks for a maximum of 6 cycles + Best supportive/best standard of care (BS/BSOC)	750	overall survival
NCT03392428	^177^Lu-PSMA-617	II	8.5 GBq (0.5 GBq per cycle) every 6 weeks	200	PSA response
NCT03780075	^177^Lu-EB-PSMA-617	I	1.11GBq (30 mCi) of 177Lu-EB-PSMA-617 1.85GBq (50 mCi) of 177Lu-EB-PSMA-617 3.70GBq (100 mCi) of 177Lu-EB-PSMA-617	30	PSA response
NCT00859781	^177^Lu-J591 + ketoconazole ^111^In-J591+ ketoconazole	II	group-1: ketoconazole + hydrocortisone followed by a single dose of 177Lu-J591 (70 mCi/m^2^) group-2: ketoconazole + hydrocortisone followed by a single dose of 111In-J591 (5 mCi)	140	proportion free of radiographically evident metastases at 18 months by CT and/or MRI scan of the abdomen and pelvis, chest X-ray, or CT scan of the chest and bone scan
NCT03545165	^177^Lu−J591 ^177^Lu−PSMA−617	I/II	cumulative 2.7 GBq/m^2^ dose of ^177^Lu−J591 and the cumulative ^177^Lu−PSMA−617 dose from 3.7 GBq to 18.5 GBq. Dose escalation in 6 different dose levels (3+3 dose−escalation study/de-escalation design)	48	dose-limiting toxicity (DLT), cumulative maximum tolerated dose (MTD), PSA response
NCT03276572	^225^Ac−J591	I	a single dose of ^225^Ac−J591 (13.3 KBq/Kg–93.3 KBq/Kg or 0.36 μCi/Kg–2.52 μCi/Kg)	42	dose-limiting toxicity (DLT)

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
