# Peer review of "Targeted Radionuclide Therapy of Prostate Cancer—From Basic Research to Clinical Perspectives"

_molecules, 2020, doi:10.3390/molecules25071743_

Round 1

Reviewer 1 Report

I have several minor comments and suggestions to the authors. 

I suggest to reformulate the article title, to sound better and more clear.

E.g.: "Targeted radionuclide therapy of prostate cancer - from basic research to clinical perspectives"

Page 1, line 24: Instead of "radioisotopes" I suggest to use radionuclides, since the review deals with various radioactive elements. 

Page 2/3, lines 67-98:  Please mention also the PSA expression in women Skene´s glands (see e.g.: 10.1006/gyno.1994.1294)

Page 4, line 137: Please check the spelling. 

Page 7, figure 3 : Please put the molecules in alphabetical order and align their structures.   

Page 8, lines 315-332 and chapter 4.3.: Please discuss the contribution of conversion electrons to the overall dose. See e.g. recent review: 10.1186/s41181-019-0075-2. Influence of various radiation types induced bystander effects should be discussed. 

Page 10, chapter 4.2. Alpha emitting radionuclides. Please discuss in more detail the nuclear recoil effect particularly in small molecules. This should be mentioned in Conclusions part as well.  

Page 11, line 450:  For completeness, I suggest to mention also the hydroxyapatite NPs and CaCO3 microparticles. See e.g.: 10.1039/C9RA08953E and 10.1002/jlcr.3610.

Several places in the text  

Please check radionuclides mass number formats. E.g. page 15, line 650, 654, 657, 667,...

Please verify and use the same notation for PSMA-617 ligand instead of e.g.: 177Lu-DKFZ-617, 177LuPSMA-617, 225Ac-PSMA-617, 225Ac-PSMA617. 

Reviewer 2 Report

This review tried to summarize the potential therapy targets for prostate cancer and wants to focus on the targeted radiotherapy. From my point of view,  this review is too lengthiness and it is hard to get the key point. Here are some  suggestions:

For the potential therapy targets, I think it might be okay (from the 1st page to 8th page). But for the targeted radiotherapy, at first, the authors need to concern what are the main point. For my concern, those details for β- particles, α-particles and Auger electrons are not necessary, the author should consider to conclude that by fewer paragraphs and list a table to summarize the similarities and differences.

In addition, the authors should not just provide a lot of examples of the PSMA targeting ligands labeled with radionuclides. I expected the review does not mean simply collecting a lot a papers and list with every case’s details. The authors need to compare those cases and give their own conclusions and prospective.

Thus, my advice is the review need to be refined. If it could combine with some meta-analysis data, it will be much better/appropriate.

Round 2

Reviewer 1 Report

All my suggestions were implemented in the article. 

I suggest to accept the paper in present form.